# Use of Electroencephalography (EEG) for the Analysis of Emotional Perception and Fear to Nightscapes

**Mintai Kim [1], SangHyun Cheon [2] and Youngeun Kang [3],***

[1] Landscape Architecture Program, Virginia Tech, Blacksburg, VA 24060, USA; mintkim@vt.edu
[2] Department of Urban Planning and Design, Hongik University, Seoul 04066, Korea; scheon@gmail.com
[3] Research Lab, Site Planning Co., Ltd., Busan 48505, Korea
**\*** Correspondence: jiyoon8936@gmail.com

**Abstract:** As the necessity for safety and aesthetic of nightscape have arisen, the importance of nightscapes (i.e., nighttime landscape) planning has garnered the attention of mainstream consciousness. Therefore, this study was to suggest the guideline for nightscape planning using electroencephalography (EEG) technology and survey for recognizing the characteristics of a nightscape. Furthermore, we verified the electroencephalography (EEG) method as a tool for landscape evaluation. Therefore, this study analyzed the change of relative alpha wave and relative beta wave and perceived fear of participants depending on twelve nightscape settings (four types of settings: Built nightscape images group with an adult; Built nightscape images groups without an adult; Nature-dominant nightscape images with an adult; and Nature-dominant nightscape images without an adult). Our findings indicate that the most fearful nightscape setting was recorded in Built nightscape images groups without an adult figure in perceived fear result depending on four types of nightscape settings. In Nature-dominant nightscape images, on the other hand, the nightscape setting with an adult figure was more fearful than the setting without an adult. The interaction effect between landscape type (built and nature-dominant) and adult presence towards perceived fear was verified and it showed that the image with adult affects landscape type. For electroencephalography (EEG) results, several brain activities in the relative alpha and beta wave showed significant differences depending on nightscape settings, which situates electroencephalography (EEG) as an invaluable tool for evaluating landscapes. Based on our physiological electroencephalography (EEG) experiment, we provide a new analytic view of the nightscape. The approach we utilized enables a deeper understanding of emotional perception and fear among human subjects by identifying the physical environment which impacts how they experience nightscapes.

**Keywords:** electroencephalography; EEG; psychophysiological responses; landscape evaluation; nightscapes; sustainable landscape design; fear; night pollution

## 1. Introduction

### 1.1. Background

As the necessity for safety and aesthetic of nightscape have arisen, the importance of nightscapes (i.e., nighttime landscape) planning has garnered the attention of mainstream consciousness. Many local governments are recognizing that well-designed nightscapes can enhance the image of a city and subsequently attract more residents, investors, and tourists. From an urban planning point of view, there is difficulty reconciling conflicts and interests between producers of light (beneficiary) and consumers, to draw consensus of the community, and to reflect these in light pollution standards and management systems. In this context, research on nightscape planning has been carried out in

terms of night pollution [1,2], tourism development [3], safety issues [4], etc. However, the preceding studies have mostly focused on a particular structure to conduct a field survey rather than on empirical data [5]. Experimental data of nightscape are significant for human health as excessive lighting can cause fatigue, serious illness such as cancer, and accidents [6].

Experiential data reflect the psychological elements of the participants. Since more than 70% of information is obtained through visual sense in human's five representative senses (sight, hearing, touch, smell, and taste), many studies [7,8] have been conducted to analyze emotions aroused from visual stimuli. Therefore, it is important to study psychological aspects among the effects of lighting on the human body, such as concentration, nervousness, and fear [9]. This study validated the relationship between nighttime environments and fear as one of the affective responses to nightscape. We examined participants' reported levels of fear that directly correspond to the interaction of lighting positions and the presence of specific physical elements in the landscape.

In this study, we developed a new method of analyzing nightscape using Mobile electroencephalography (EEG), which is directly related to people's perception of the environment. The existing studies do not directly evaluate the EEG response to nightscape in combination with a survey analysis to assess human perception. Recent laboratory-based neuroimaging studies indicate that various environments may be associated with characteristic patterns of brain activity [10–12]. Mobile EEG provides a non-invasive way to capture emotional states of human research subjects. Furthermore, research that utilizes Mobile EEG requires rigorously controlled experiments and complex analytical tools. Mobile EEG is increasingly being used beyond the clinical and experimental environments; it is now frequently used to monitor brain function and cognition in real life situations [13]. A unique aspect of Mobile EEG is its ability to gather the participants' response data on a second-by-second timescale with virtually no interruptions [14]. Recent Mobile EEG research shows how people can evaluate, visualize, explore, and develop a spatial perception of architectural designs [15].

The purpose of this study was to suggest guideline for nightscape planning using EEG technology and survey for recognized characteristics of a nightscape. Furthermore, we verified the EEG method as a tool for landscape evaluation. We used survey methods to investigate participants' subjective perception of fear level to help interpret EEG data in a real-world setting by using mobile EEG apparatus. While EEG output provides a real-time psychophysiological measurement of response to changing environments, self-reporting of fear provides a context and understanding of these changes.

*1.2. Studies on Nightscape and Desirable Landscape Types for Nightscape Studies*

Several previous studies on landscape perception have been associated with measuring how people perceive specific surrounding environmental settings during the daytime. Most of these studies have derived design guidelines following their findings. Nighttime design guidelines, however, for a particular environmental setting have not been as well developed as nightscape perception research. Lee et al. [16] analyzed subjective characteristics of light in nightscapes and studied the relationship between lighting design and people's perceptions of nightscapes. Ahn et al. [17] attempted to evaluate nightscapes by identifying variables that affect people's perception of nighttime streetscapes. Park et al. [18] studied the maintenance and improvement of nightscapes through field surveys. Most of these studies use qualitative methods.

Research has discussed the interplay between landscape types and the physiological response of human beings [19]; it is very critical to divide landscape types in landscape evaluation studies. It is common to divide by dichotomy, e.g., natural versus built landscape, in existing studies [14,15,20–22], but there have been various ways to divide landscape types in previous studies. Ulrich et al. [23] divided landscape types into six: plant environment including trees and other vegetation; water environment, primarily flowing water and that which involved trees; congested traffic; normal traffic; crowded pedestrian environment; and common pedestrian environment. Chang et al. [10] divided landscape types more specifically depending on the wildness level: extensive landscape such as mountain, small landscape such as Japanese gardens, and abstract landscape such as a view from

window. Similarly, landscapes from daytime can be divided in various ways, because people can perceive their detailed differences. However, landscape type from nighttime (nightscape) should be differently considered when it comes to arousing fear and its observability. Fisher and Nasar [24] argued that daytime environments such as tree can increase fear at night because it provides concealment, limited prospects, and blocked escape routes. Moreover, the detailed landscape types in landscape evaluation research make it difficult for people to distinguish landscapes.

Therefore, the specific landscape types in this study were divided into natural and built landscape including buildings, low free-standing walls, tall and short trees, and shrubs. Since there were few nightscape scenes without any built elements, we set the images including mostly natural elements as Nature-dominant nightscape. Additionally, we investigated the effects of the presence of a human figure in a nightscape, because the presence of a stranger in a nighttime landscape is suspected to elicit fear.

## 1.3. Studies on EEG

EEG has been used as a tool to supplement surveys or experts' opinion that have been commonly utilized in landscape evaluation field. Recent studies using neuroimaging methods in environmental psychology have shown that different types of urban environments interact differently with varying environments in relation to the distinctive patterns of brain activity [14]. Existing studies using EEG in this way have explored how people perceive different environment settings, and these studies [10,14,15] mainly compare the natural landscape versus built landscape among various settings (see details in Table 1). For example, Roe et al. [15] investigated EEG how the brain engages with natural versus urban setting, suggesting that natural based landscapes are associated with greater levels of meditation and lower arousal than urban scenes. Tilley et al. [14] measured the level of excitement, engagement, and frustration using EEG depending on specific urban and natural settings (eight types of environmental settings). Tilley et al. also proposed a detailed design implication that compares EEG results with different settings.

As presented above, differences in perceived color [25], fractal pattern [26], and biodiversity [27,28] as well as differences in brain activities by landscape type have been discussed.

Kim and Lee [25] used EEG to derive a design implication that alpha wave can be used to create a peaceful space for alpha sound and to create lively spaces using beta waves. Here, the various brain wave such as alpha and beta wave are used to evaluate brain activity by proxy measurements. The measurement of brain activity can be divided into four types in general: delta (<4 Hz) features slow and loud brainwaves and is generated in deepest meditation and dreamless sleep; theta (4–7 Hz) occurs most often in light sleep or extreme relaxation; alpha (8–13 Hz) is dominant during quietly flowing thoughts and in some meditative states; and beta (14–30 Hz), which dominates our normal waking state of consciousness when attention, is directed towards cognitive tasks [29].

As the recent EEG technology develops, the use of mobile EEG has been widespread in related studies, and new approaches combining different methodologies such as eye tracking [30], electromyography and blood volume pulse [10], and in-depth interview [14] with EEG are also increasing to validate EEG's effectiveness. In addition to EEG technology, fMRI (functional Magnetic Resonance Imaging), another technology for measuring brain activity, has been used to compare landscape characteristics in other studies [19,31]. Kim et al. [31] used functional MRI in response to viewing rural and urban living environment, which suggested an inherent preference toward nature-friendly environment. Tang et al. [19] compared the restorative value of four types of landscape environments (urban, mountain, forest, and water) using questionnaires and fMRI as well, and found the water type was the most restorative environment among other stimuli.

Many EEG studies in aspects of environment have engaged with showing generally beneficial effects of green spaces or specific colors and environments in deriving preference or restorative effects from natural landscape. However, there is no research regarding its beneficial effect on nightscape. Accordingly, this study used EEG to evaluate nightscapes related with its fear and

settings (nature-dominant versus built landscape). Not only these landscape type but also appearance of an adult in each image was compared to verify EEG's usability in landscape evaluation field.

**Table 1.** Related research and its experimental environments.

| Reference | Experimental Settings | Used Brain Waves |
|---|---|---|
| [10] | Wildness landscape (Extensive landscape, small environment, and abstract landscape) | Alpha |
| [25] | Emotional color settings | Alpha and beta |
| [15] | Landscape and urban scenes for the restorative potential | Alpha, beta, delta, and theta |
| [27] | Various deciduous broad-leaf forest | Alpha, beta, delta, and theta |
| [32] | Varying locations and vegetation density in natural landscape | Alpha |
| [14] | Built urban environment and an urban green space environment (eight different settings) | Levels of excitement, engagement, and frustration (as interpreted by proprietary EEG software) |

*1.4. Research Hypotheses*

Based on the background and literature review, we investigates nightscape characteristics comparing EEG data and reported level of fear for suggesting nightscape guideline. The specific research hypotheses corresponding to the objectivity of this study are below.

**Hypotheses 1 (H1).** *There is an interaction between landscape type and presence of a person toward perceived fear and EEG.*

**Hypotheses 2 (H2).** *People's level of fear varies depending on presence of a person and landscape type.*

**Hypotheses 3 (H3).** *The relative alpha and beta waves of EEG vary depending on presence of a person and landscape type.*

## 2. Methods and Data

*2.1. Research Process*

The method of this study was divided into three sections: selection of experimental images, experiments, and analysis of EEG results and survey (Figure 1). First, we selected the experimental images for the study (twelve images divided into two types: nature-dominant nightscape and built nightscape) with consultation from five experts. Second, EEG and perceived fear depending on nightscape settings were collected and evaluated simultaneously by each participant. Third, we analyzed the EEG result and survey (perceived fear). We performed ANOVA test to compare the differences of EEG and perceived fear depending on nightscape settings. After going through these three steps, we reviewed the results of the study, which suggests the implication for nightscape planning and utility of EEG.

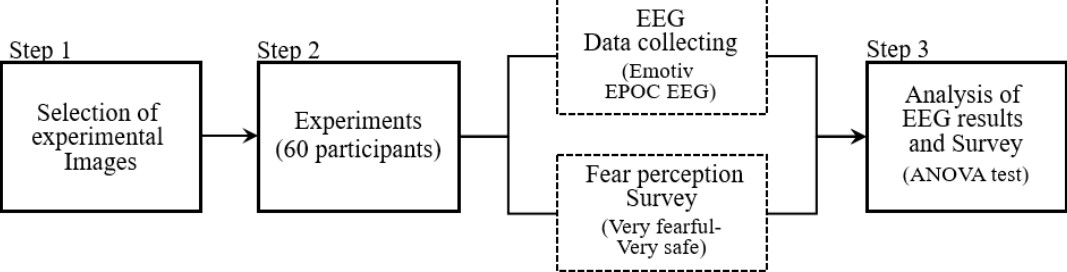

**Figure 1.** Research process of this study.

## 2.2. Participants

A total of 60 students, professors and staff from various departments at Virginia Tech participated in this study. Based on Jazi et al. [33], precautions for this study (absence of any cardiovascular or neurological disorder or metallic implant, no potential chances of pregnancy, no consumption of street drugs, and refraining from coffee/alcohol/nicotine intake 24 h prior to testing) were informed to participants before the experiment. They were assigned randomly to one of two groups: built nightscape image group (BNIG, n = 30), and nature-dominant nightscape image group (NNIG, n = 30). Among them, 32 were men and 28 were women. Participant' age ranged from 20 to 40 (53.3% were in their twenties, 28.3% in their thirties, and 18.3% in their forties). Our research protocol and survey instrument were approved by the Institutional Review Board of Virginia Tech.

## 2.3. Experimental Images

To verify these assumptions, twelve digital photographs were used to conduct surveys at the same time as the EEG experiments. There was a discussion about the elicitation work to select these photo settings with five experts who are professors researching in landscape architecture, architecture, and urban planning.

The six sets of photos used in this study were taken during the same season at the Virginia Tech campus. We identified two core environments of nightscape each with three photographs: "built" (or "grey") scenes (i.e., buildings, roads, walls, etc.) as the built nightscape images and "green" elements (fields, forest, and parkland) as the nature-dominant nightscape images. In addition, each set had two photos, one with an adult figure and another one without an adult figure. As participants viewed each image, they were asked to rate the level of fear elicited by the nightscape on a seven-point Likert-type scale (where 1 = very safe, and 7 = very fearful). The examples of the experimental images are below (Figure 2), and all of stimuli used in this study are depicted in Appendix A.

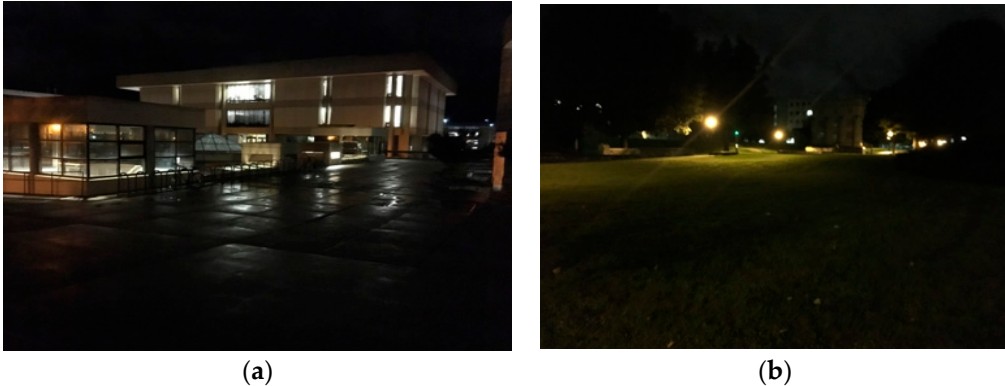

|(**a**)|(**b**)|

**Figure 2.** The examples of experimental images taken by the authors: (**a**) built scene without an adult; and (**b**) nature-dominant scene without an adult.

## 2.4. Apparatus (Emotiv EPOC EEG Device)

We selected the Emotiv EPOC EEG device in this study (see Figure 3). The Emotiv Epoc headset was used to extract the EEG data from each participant. Visual stimuli were presented on a 19-inch LCD monitor. Using the Emotiv Test Bench and OpenVibe as software, we captured the raw EEG output coming from the headset. This headset has 14 electrodes (saline sensors) that take readings from activation sites on the surface of the brain, and comes with a suite of software packages. It also includes a two-axis gyroscope to detect the wearer's head motion and orientation (see details in Figure 4).

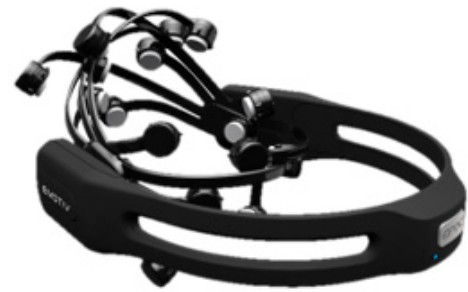

**Figure 3.** Emotiv EPOC EEG device used in this study. Reference: [34]

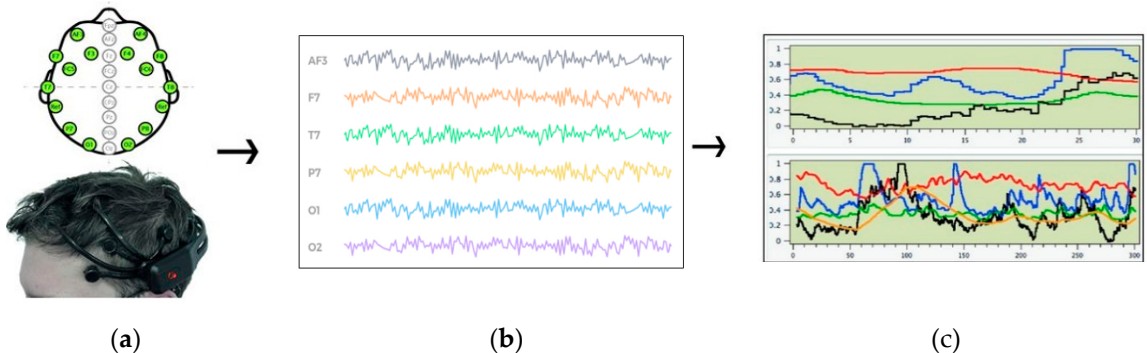

| (**a**) | (**b**) | (c) |

**Figure 4.** Process of collecting EEG data: Emotiv EPOC records EEG signals from 14 sensors position according to the 10–20 international system: (**a**) raw EEG (the electrodes location) signals were "translated" and classified in four different emotional states; (**b**) output from Emotiv; and (**c**) output sample using Testbench software from Emotiv Control Panel and Affective suite (EEG data belong to the authors). Reference: [34,35]

## 2.5. Measurements (EEG)

To remove the residuals from the EEG original data, we performed Fast Fourier Transform (FFT) after filtering and then conducted PSA (Power Spectrum Analysis). From this step, the absolute power value and the relative power value for each frequency were derived. The relative power value means a power value equal to an absolute value difference between individuals. This represents the sum ratio of the frequency set for the total sum of the entire frequency ranges in the power spectrum. Previous studies have indicated that the EEG signal may be different for individuals and environments. That is, even when the external conditions such as temperature and brightness are measured in the same way, the electric resistance varies depending on the state of the scalp and the state of the mental state, so the result of EEG may be different.

Among 12 channels of EEG, we used the main eight channels from frontal (Fp1, Fp2, F3, and F4), and occipital (O1 and O2) and parietal (C3 and C4) to capture two main waves: alpha and beta waves (see details in Figure 5). The alpha wave (8–12.99 Hz) appearing when relaxing [6,28] and the beta wave (13–29.99 Hz) appearing when being anxious or stressed [11,21,23] were extracted and analyzed among various types of brain waves. We also used and analyzed the relative wave, which is the whole interval of the alpha and beta wave, to determine the EEG differences between the participants.

## 2.6. Statistics

All data were analyzed using SPSS 15.0. Two-way *ANOVA* was carried out to verify fear results and changes by frequency ranges of EEG depending on different environment settings (depending on adult presence and landscape type). If no interaction between two factors was found, we performed a one-way ANOVA to compare differences between groups including post-hoc analysis (Scheffe).

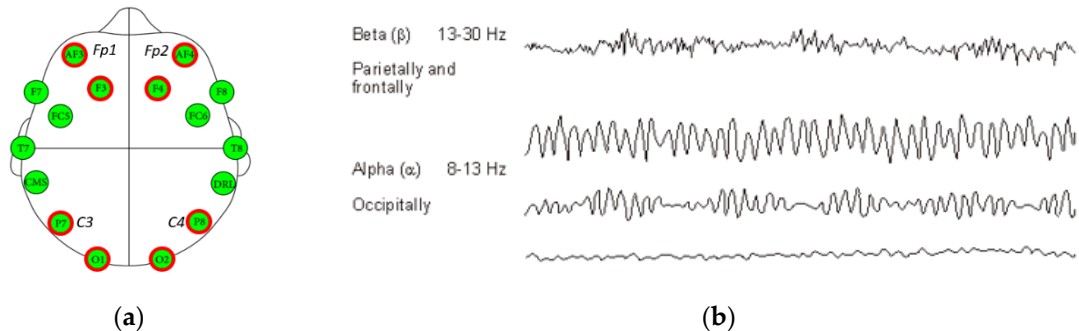

(**a**)
(**b**)

**Figure 5.** EEG measurements: (**a**) the main eight EEG areas used in this study (marked with a red boundary); and (**b**) EEG rhythms showing the frequency of alpha wave (bottom in the figure) and beta wave (above in the figure). Reference: [34,36]

## 3. Results

### 3.1. Self-Reported Level of Fear

The results of the fear rating for each nightscape image is as follows (Table 2). Compared with the mean between two groups, the level of fear tended to be higher in BNIG. BNI without adult figure rated the highest fear among four types of landscape images. On the other hand, the lowest fear was in NNI without adult figure.

**Table 2.** Self-reported level of fear depending on adult presence and landscape type.

| Adult Presence | Landscape Type | Mean | Std. | N |
|---|---|---|---|---|
| | BNI | 5.43 | 0.54 | 30 |
| Without adult | NNI | 4.44 | 0.72 | 30 |
| | Total | 4.94 | 0.63 | 60 |
| | BNI | 4.71 | 0.58 | 30 |
| With adult | NNI | 4.87 | 0.57 | 30 |
| | Total | 4.79 | 0.57 | 60 |
| | BNI | 4.94 | 0.65 | 60 |
| Total | NNI | 4.79 | 0.58 | 60 |
| | Total | 4.87 | 0.62 | 120 |

BNI, Urban Nighttime Image, NNI, Landscape Nighttime Image. Homogeneity was verified by Levene's test of equality ($p = 0.683$).

Table 3 shows the results of two-way ANOVA, which suggested the significant differences toward perceived fear in adult presence and landscape type. The group differences in landscape type were found to be significant ($p = 0.00$), but adult presence was not significant ($p = 0.187$). In addition, the interaction effect between two factors (landscape type and adult presence) was significant, which means H1 in this study was verified. In other words, adult presence can affect the perceived fear depending on landscape type, and the opposite effect (landscape type toward perceived fear depending on adult presence) can be interpreted in the same way.

Since the interaction effect between two factors was verified in the previous analysis, we performed post-hoc to figure out which condition on each factor can affect perceived fear (Table 4 and Figure 6). The results show that landscape without adult affects perceived fear on landscape type ($p = 0.000$). On the other hand, when there is an adult, it does not affect the result on landscape type ($p = 0.289$).

In sum, it was found that perceived fear in nature-dominant was generally lower than built landscape, however adult presence affects the perceived fear. Specifically, landscape without adult made people feel more fearful in built nightscape, but, in nature-dominant nightscapes, people feel less fearful in nightscape without adult.

**Table 3.** The result of ANOVA (test of between subject effects).

| Division | SS | df | MS | F | P |
|---|---|---|---|---|---|
| Corrected Model | 15.71 * | 3 | 5.23 | 13.68 | 0.000 ** |
| Intercept | 2838.89 | 1 | 2838.89 | 7.42 | 0.000 ** |
| Adult Presence | 0.68 | 1 | 0.68 | 1.76 | 0.187 |
| Landscape Type | 5.21 | 1 | 5.21 | 13.61 | 0.000 ** |
| Adult Presence * Landscape Type | 9.82 | 1 | 9.82 | 25.66 | 0.000 ** |
| Error | 44.04 | 116 | 0.38 | | |
| Total | 2899.00 | 120 | | | |
| Corrected Total | 60.11 | 119 | | | |

R Squared = 0.261; * $p < 0.05$; ** $p < 0.01$.

**Table 4.** Analysis of variance (post-hoc test).

| Source of Variation | SS | df | MS | F | P |
|---|---|---|---|---|---|
| Within Residual | 44.40 | 116 | 0.38 | | |
| Landscape type within Adult 1 (Without Adult) | 14.67 | 1 | 14.67 | 38.07 | 0.000 ** |
| Landscape type within Adult 2 (With Adult) | 0.36 | 1 | 0.36 | 0.94 | 0.334 |
| Adult Presence | 0.68 | 1 | 0.68 | 1.76 | 0.187 |
| Model | 15.71 | 3 | 5.24 | 13.68 | 0.000 ** |
| Total | 60.11 | 119 | 0.51 | | |

** $p < 0.01$.

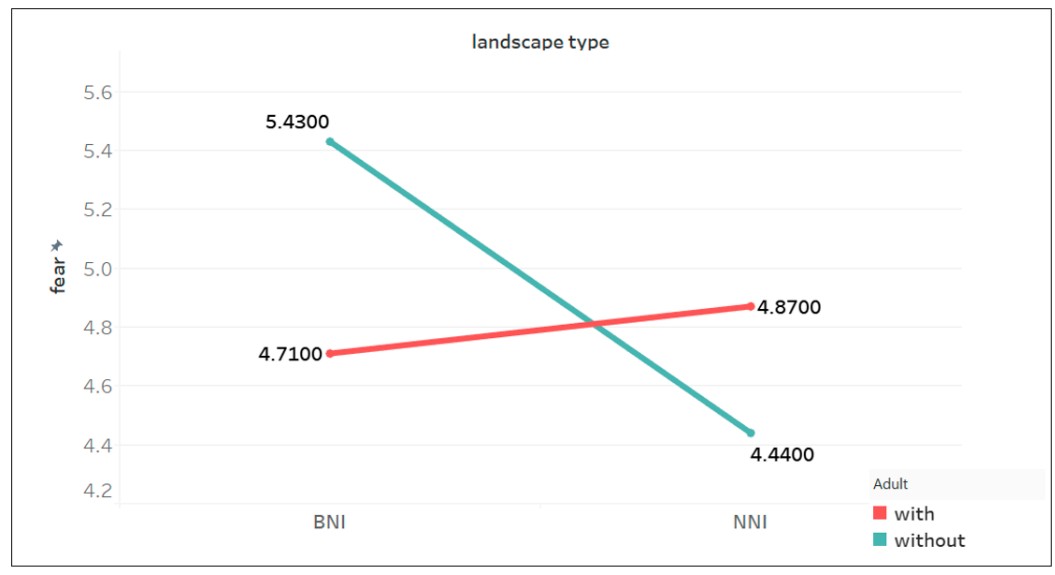

**Figure 6.** Estimated marginal means of fear (using Tableau software).

*3.2. Changes in EEG*

3.2.1. Comparison between Groups for EEG on Relative Alpha Wave

For alpha wave, one-way ANOVA between four groups (BNIG/wo, BNIG/w, NNIG/wo, and NNIG/w) was performed because the interaction effect between landscape type and adult presence toward alpha wave was not significant.

The averages of relative alpha wave were compared with eight types electrode (see Table 5). The results of BNIG showed alpha wave ratio increased in Fp1 (0.17→0.35), Fp2 (0.11→0.13), F3 (0.23→0.31), F4 (0.23→0.33), C3 (0.24→0.34), C4 (0.31→0.35), O1 (0.26→0.27), and O2 (0.19→0.27) after seeing figure including adult. The ratio of NNIG also decreased over the whole electrode areas (Fp1 (0.33→0.24), Fp2 (0.34→0.28), F3 (0.38→0.35), F4 (0.39→0.31), C3 (0.39→0.35), C4 (0.39→0.29), O1 (0.30→0.23), and O2 (0.26→0.24))

**Table 5.** Comparison between groups for EEG on relative alpha wave.

| | BNIG (n = 30) | | NNIG (n = 30) | |
|---|---|---|---|---|
| | **Without Adult Figure** | **With Adult Figure** | **Without Adult Figure** | **With Adult Figure** |
| Fp1 | 0.17 ± 0.09 | 0.35 ± 0.08 | 0.33 ± 0.12 | 0.24 ± 0.06 |
| Fp2 | 0.11 ± 0.03 | 0.13 ± 0.04 | 0.34 ± 0.12 | 0.28 ± 0.09 |
| F3 | 0.23 ± 0.07 | 0.31 ± 0.13 | 0.38 ± 0.09 | 0.35 ± 0.12 |
| F4 | 0.23 ± 0.11 | 0.33 ± 0.14 | 0.39 ± 0.09 | 0.31 ± 0.13 |
| C3 | 0.24 ± 0.10 | 0.34 ± 0.15 | 0.39 ± 0.09 | 0.35 ± 0.13 |
| C4 | 0.31 ± 0.16 | 0.35 ± 0.12 | 0.39 ± 0.10 | 0.29 ± 0.12 |
| O1 | 0.26 ± 0.10 | 0.27 ± 0.11 | 0.30 ± 0.04 | 0.23 ± 0.06 |
| O2 | 0.19 ± 0.09 | 0.27 ± 0.12 | 0.26 ± 0.10 | 0.24 ± 0.08 |

Values are presented as mean ± SD.

Table 6 shows the ANOVA for alpha wave by each electrode. The results describe that there were significant differences on every electrode (Fp1, Fp2, F3, F4, C3, C4, O1 and O2) depending on landscape types. Scheffe's post doc explains which specific groups on each electrode were statistically different. Especially, Built nightscape image group without adult (BNIG/wo) type was mostly lower than other electrodes. Specific significant differences on Scheffe's post hoc are shown on the right side of Table 6.

**Table 6.** The result of ANOVA for relative alpha wave depending on landscape types.

| Electrode | F | Sig | Scheffe's Post hoc |
|---|---|---|---|
| Fp1 | 23.29 | 0.000 ** | BNIG/wo < BNIG/w **, BNIG/wo < NNIG/wo **, BNIG/wo < NNIG/w *, BNIG/w < NNIG/w **, NNIG/wo < NNIG/w * |
| Fp2 | 55.72 | 0.000 ** | BNIG/wo < NNIG/wo **, BNIG/wo < NNIG/w **, BNIG/w < NNIG/wo **, BNIG/w < NNIG/w ** |
| F3 | 14.03 | 0.000 ** | BNIG/wo < BNIG/w *, BNIG/wo < NNIG/wo **, BNIG/wo < NNIG/w ** |
| F4 | 7.24 | 0.000 ** | BNIG/wo < NNIG/wo ** |
| C3 | 7.40 | 0.000 ** | BNIG/wo < BNIG/w *, BNIG/wo < NNIG/wo **, BNIG/wo < NNIG/w * |
| C4 | 3.61 | 0.015 * | NNIG/w < NNIG/wo * |
| O1 | 3.30 | 0.023 * | NNIG/w < NNIG/wo * |
| O2 | 3.25 | 0.024 * | BNIG/wo < BNIG/w * |

** $p < 0.01$, * $p < 0.05$; Only significant results on post hoc are displayed; /wo indicates without an adult and /w indicates with an adult.

### 3.2.2. Comparison between Groups for EEG on Relative Beta Wave (unit: mV)

For beta wave, one-way ANOVA between four groups (BNIG/wo, BNIG/w, NNIG/wo, and NNIG/w) was performed because the interaction effect between landscape type and adult presence toward beta wave was not significant similar to the alpha wave result.

The mean and standard deviation of the relative beta wave by eight EEG areas are shown in Table 7. We focused on the difference between before and after an adult appearance by two different landscape settings. The results of BNIG showed most alpha wave ratio increased in Fp1 (0.59→0.66), Fp2 (0.66→0.68), F3 (0.36→0.37), F4 (0.38→0.43), O1 (0.30→0.33), and O2 (0.29→0.31) except for C3 (0.27→0.23) and C4 (0.24→0.23) after seeing figure including adult. The ratio of NNIG increased over the whole electrode areas (Fp1 (0.58→0.72), Fp2 (0.57→0.71), F3 (0.34→0.39), F4 (0.33→0.42), C3 (0.17→0.23), C4 (0.14→0.21), O1 (0.27→0.28), and O2 (0.27→0.30)) after an adult appearance.

The result of ANOVA for beta wave by each electrode is depicted in Table 8. Unlike alpha wave's ANOVA test, statistical significance was relatively low. For Fp1, Fp2, C3 and C4, there was significant difference depending on landscape types. The detailed results of the post hoc are as follows.

**Table 7.** Comparison between groups for EEG on relative beta wave.

| | BNIG (n = 20) | | NNIG (n = 20) | |
|---|---|---|---|---|
| | Without adult Figure | With Adult Figure | Without Adult Figure | With Adult Figure |
| Fp1 | 0.59 ± 0.14 | 0.66 ± 0.08 | 0.58 ± 0.15 | 0.72 ± 0.08 |
| Fp2 | 0.66 ± 0.06 | 0.68 ± 0.09 | 0.57 ± 0.15 | 0.71 ± 0.09 |
| F3 | 0.36 ± 0.09 | 0.37 ± 0.10 | 0.34 ± 0.12 | 0.39 ± 0.12 |
| F4 | 0.38 ± 0.13 | 0.43 ± 0.11 | 0.33 ± 0.16 | 0.42 ± 0.11 |
| C3 | 0.27 ± 0.15 | 0.23 ± 0.08 | 0.17 ± 0.07 | 0.23 ± 0.09 |
| C4 | 0.24 ± 0.11 | 0.23 ± 0.10 | 0.14 ± 0.08 | 0.21 ± 0.09 |
| O1 | 0.30 ± 0.10 | 0.33 ± 0.09 | 0.27 ± 0.12 | 0.28 ± 0.05 |
| O2 | 0.29 ± 0.11 | 0.31 ± 0.11 | 0.27 ± 0.14 | 0.30 ± 0.06 |

Values are presented as mean ± SD.

**Table 8.** The result of ANOVA for relative beta wave depending on landscape types.

| Electrode | F | Sig | Scheffe's Post hoc |
|---|---|---|---|
| Fp1 | 8.83 | 0.000 ** | BNIG/wo < NNIG/w **, NNIG/wo < NNIG/w ** |
| Fp2 | 9.84 | 0.000 ** | BNIG/wo < NNIG/wo *, BNIG/w < NNIG/wo **, NNIG/w < NNIG/wo **, |
| F3 | 1.16 | 0.329 | - |
| F4 | 3.22 | 0.052 | - |
| C3 | 4.15 | 0.008 ** | BNIG/wo < NNIG/wo ** |
| C4 | 6.25 | 0.001 ** | BNIG/wo < NNIG/wo **, BNIG/w < NNIG/wo ** |
| O1 | 1.88 | 0.136 | - |
| O2 | 0.42 | 0.738 | - |

** $p < 0.01$, * $p < 0.05$; Only significant results on post hoc are displayed; /wo indicates without an adult and /w indicates with an adult.

Figure 7 shows the general comparison depending on four landscape settings by brain wave (alpha and beta waves). In alpha wave, the dispersion between eight electrodes was relatively smaller than the beta wave. NNIG/wo in alpha wave has the highest value, and overall NNIG value is higher than beta wave. On the other hand, the comparison of beta wave depending landscape types shows that the appearance of an adult tended to be more influential than the landscape element (i.e., natural element and built element).

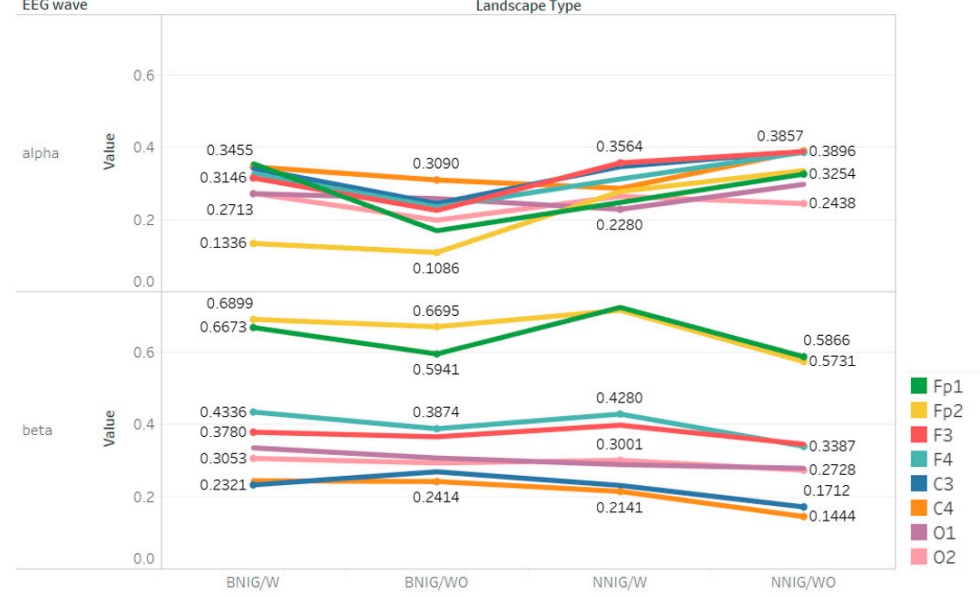

**Figure 7.** Comparison between groups for EEG on relative alpha and beta waves; /wo indicates without an adult and /w indicates with an adult (using Tableau software).

## 4. Discussion

*4.1. Perceived Fear Differences Depending on Settings and Usability of EEG in Landscape Evaluation*

This study analyzed the relationship between EEG and fear dependent upon various nightscape settings. We analyzed the relative alpha and beta waves depending on four types of nightscape settings including interpreting recorded fear on each nightscape settings from 40 participants. We focused not only on different settings depending on landscape type (BNIG and NNIG) and adult presence, but also the interaction between these two factors. In EEG part, the reasons we used the relative alpha and beta waves among various types of EEG wave was that alpha is known to occur when one is feeling stable and relaxed while beta is known to occur when one is concentrating. Therefore, it was assumed that there would be a negative relationship between fear and alpha wave and positive relationship between fear and beta wave. We also assumed that the alpha and beta waves would vary depending on the presence of an adult in each nightscape setting. The results of this study are summarized as follows and the three hypotheses we set were all verified.

First, our results show that the most fearful nightscape setting was recorded in BNIG without the adult figure when comparing self-recorded fear depending on four types of nightscape settings. In NNIG, on the other hand, the nightscape setting with adult figure was more fearful than the nightscape setting without adult. The critical part of the perceived fear difference depending on settings is the significant interaction between landscape type and adult presence, which are two main independent variables towards perceived fear. That is, adult presence in landscape settings can affect perceived fear. Specifically, the difference between built nightscape and nature-oriented nightscape was evident in images without adult. When people perceive nightscape, it is understood that adult presence make people disperse their gaze, which further implies the importance of setting conditions in the landscape evaluation.

Second, overall EEG wave (eight brain areas in alpha and beta waves) was affected by not only nightscape type, but also the presence of an adult. Especially, the EEG response in frontal lobes, which is related to the cognitive function, showed a significant relationship between the self-reported fear. The result of relative alpha wave indicated that there was a significant difference in Fp1, F3, and O3 brain areas according to a presence of adult. This means the relative alpha wave is affected by the presence of people. The result of Fp2 showed there are clearly differences if the setting is built or nature-dominant. All brain activity was increased in NNIG compared to BNIG when only comparing settings. As reported, the alpha wave increased primarily when the test subject felt relaxed. Hence, decreased alpha wave values mean that the brain has changed to a tension and excitement state, thus this can be quite related to state feels fear. This is consistent with the self-reported fear in which fear level decreased with an adult figure on BNIG and increased fear level with an adult on NNIG. Several brain activities in the relative beta wave including Fp1, Fp2, and C4 showed significant differences. Specifically, the differences in Fp1 showed BNIC/wo was lower than NNIG/w and NNIG/wo was lower than NNIG/w, which means the setting as well as the presence of an adult affected people's brain activity. Overall result on beta wave indicated that, if there was an adult in setting, the relative beta wave increased. This implies there is no direct relationship between beta wave result and self-reported fear. The beta wave is generally divided into slow beta wave (13–21 Hz) and fast beta wave (22–30 Hz). Beta wave commonly increased during the task requiring attention compared to the relaxed state, and activated beta wave reflects an increase in cognitive function due to high intensity information processing activities. Accordingly, it is supposed that increasing beta wave in setting with an adult tells people consciously judges they can be threatened by an adult in nightscape setting. We found that beta wave increases when paying more attention, while alpha wave decreases depending on nightscape type in this study, and this result is consistent with previous research [15]. EEG research using image-based EEG is as effective as engaging participants in a real environment.

*4.2. Nightscape Design*

There have been very few studies regarding nightscape design while daytime landscape design studies [37] continue to be analyzed. Studies related to existing nightscape studies have been mainly focused on light itself [38,39] or images on nightscapes [40,41]. The nightscape, complete with awe-inspiring atmospheric events and potentially restorative fascinating stellar views, requires more empirical investigation [42]. Nightscape design is closely related to preference, satisfaction, and light pollution as well as perceived fear. Therefore, we invite other analysts in the field of nightscape design to extend our findings. The insight obtained in this study regarding nightscape design is that green element such as parks, shrubs, trees, flowers, etc. function to reduce fear and facilitate relaxation more than built elements. It is also important to consider the significant differences between nightscape settings through EEG, which implies its usability in nightscape study, especially for nightscape design. Beyond the experiment in this study, constant communication between landscape designer and people perceiving the environment at night is significant. The physical environment settings for improving usability at night can be further improved by seizing which parts are more fearful or pleasing. Recent research presents the possibility to measure nightscape using sophisticated technology (e.g., airborne hyperspectral cameras [38]). In sum, various studies comparing perceived nightscape and measured nightscape by various tools present new possibilities for enhancing the quality of nightscapes.

## 5. Conclusions

This study analyzed perceived fear with EEG focusing on the changing alpha and beta waves of participants in four different types of nightscape settings to suggest its usability in nightscape design. Our findings indicate the corresponding measures of fear vary according to the environmental settings, which are described as follows: (1) the interaction between landscape type and adult presence was verified, which means that other conditions such as adult presence besides landscape settings can affect landscape evaluation; (2) the perceived fear depending on the four settings was statistically different and the most fearful nightscape setting was BNI without the presence of an adult; and (3) the differences of the alpha and beta waves depending on settings were significant, which means EEG can be one of the measures for evaluating nightscape characteristics (e.g., fear, preference, etc.). The alpha wave recorded was relatively high in nightscape settings consisting of natural elements. Additionally, the presence of an adult affects the brain wave (both alpha and beta waves) regardless of the nightscape setting.

This study has limitations due to the relatively few landscape types investigated and comparatively low number of participants. We posit that this could be extended in future studies. However, the approach we employed enables a deeper understanding of the emotional perception and fear among human subjects by identifying the physical environment, which impacts how they experience nightscapes. Although more specific nightscape settings should be compared using EEG in future studies, our findings based on the physiological EEG experiment provide a new analytic approach to studying nightscapes.

**Author Contributions:** All authors have contributed to the intellectual content of this paper. The first author, M.K., developed the flow of this study and wrote most of the manuscript. He was also responsible for all statistical analysis including EEG analysis and group differences. S.C. contributed to discussion part for suggesting nightscape design. Y.K. substantially contributed to the research design, wrote some of the manuscript and contributed to interpretation of all results and discussion.

**Funding:** This research was supported by a grant (Grant 17CTAP-C129890-01) from Land, Infrastructure and Transportation R&D Program (Science Technology Promotion Research Project) funded by Ministry of Land, Infrastructure and Transport of Korean government. The publication cost of this work was supported by the Virginia Tech Open Access Subvention.

**Conflicts of Interest:** The authors declare no conflict of interest.

## Appendix A

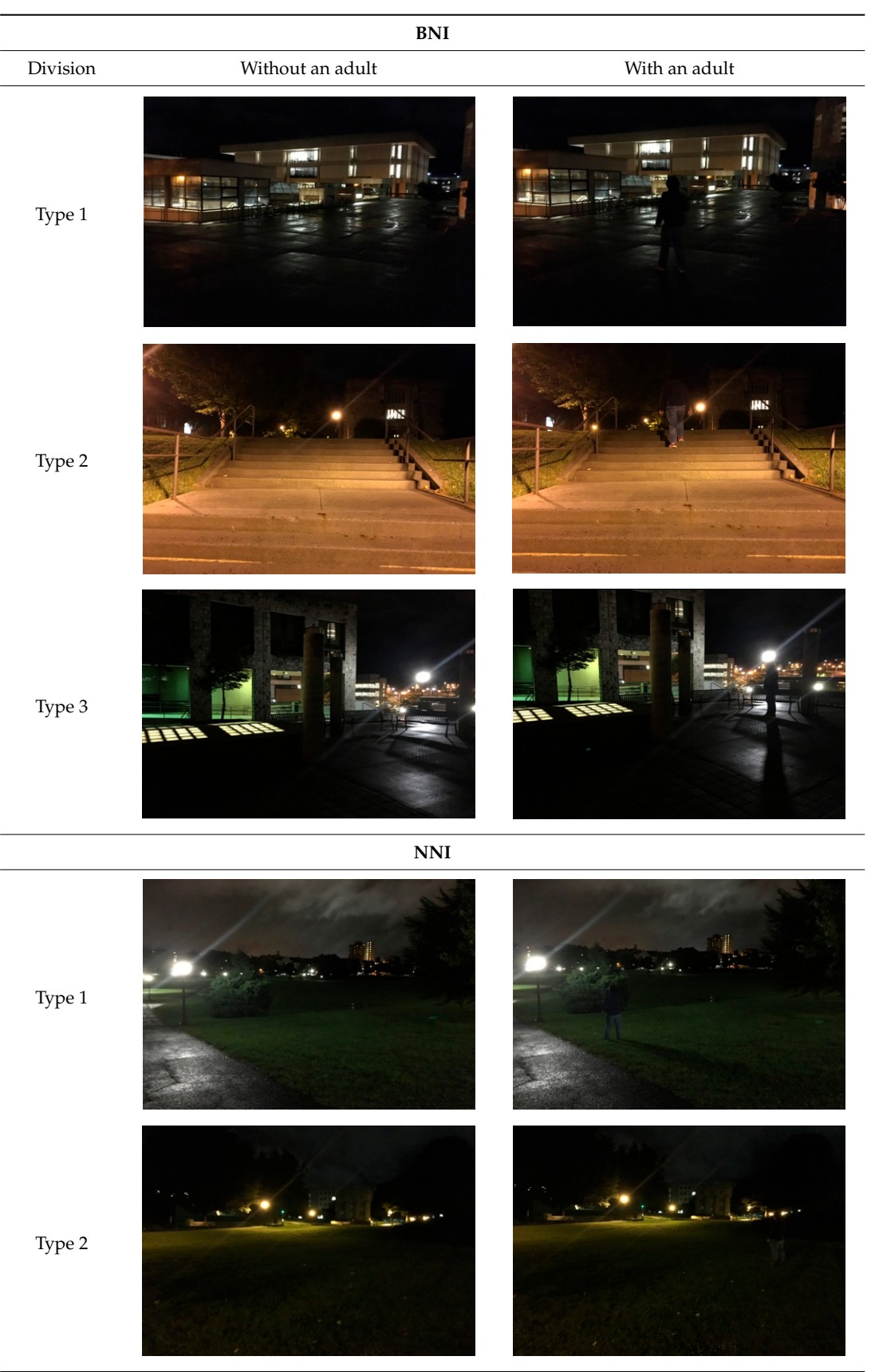

| | | |
|---|---|---|
| Type 3 | 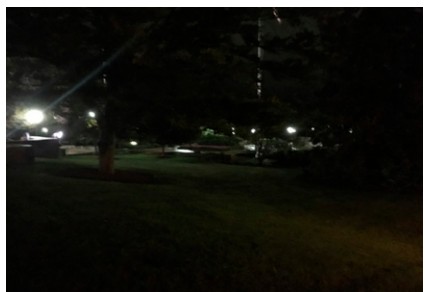 | 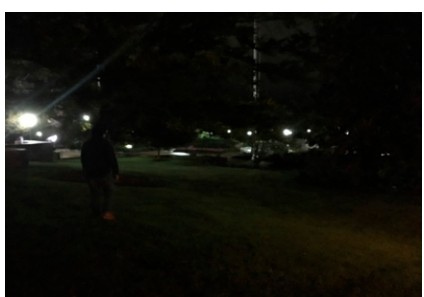 |

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
