# Peer review of "Use of Electroencephalography (EEG) for the Analysis of Emotional Perception and Fear to Nightscapes"

_sustainability, doi:10.3390/su11010233_

Round 1

Reviewer 1 Report

The authors have further improved the paper and this study and its findings would help drive the guideline for nightscape planning using electroencephalography (EEG) technology and survey for recognizing the characteristics of a nightscape.

Author Response

First of all, we are very grateful for the detailed review of our paper. We believe your comments will greatly contribute to enhancing the completeness of our manuscript, what we modified and additional comments are listed below in order.

Revised parts are marked red in manuscript file.

We appreciate that you did a detailed evaluation of our paper.

Reviewer 2 Report

The paper is very interesting an the used methods are well explained.

Still the used method considering the sample size turn the conclusions very week.

IN this regard, as mentioned on my previous review, the sample size should be larger and new input data is needed.

Author Response

First of all, we are very grateful for the detailed review of our paper. We believe your comments will greatly contribute to enhancing the completeness of our manuscript, what we modified and additional comments are listed below in order.

Revised parts are marked red in manuscript file.

We appreciate that you did a detailed evaluation of our paper.

Still the used method considering the sample size turn the conclusions very week.

-> As presented in the previous comments, comparatively low sample in EEG study was perceived as valid according to previous studies below. It is also found out the around 40 participants in EEG study is quite higher than previous study. In order to improve our manuscript, we reanalyzed the data and found out other aspect of result and implication.

* Jazi, S.D.; Modolo, J.; Baker, C.; Villard, S.; Legros, A. Effects of A 60 Hz Magnetic field of up to 50 milliTesla on human tremor and EEG: A pilot study. International Journal of Environmental Research and Public Health. 2017, 14, 1446.

* Tang, I.; Tsai, Y.; Lin, Y., Chen, J.; Hsieh, C.; Hung, S.; Sullivan, W.C.; Tang, H.; Chang, C. Using functional Magnetic Resonance Imaging (fMRI) to analyze brain region activity when viewing landscapes. Landscape and Urban Planning. 2017, 162, 137-144.

* Kim, T.H.; Jeong, G.W.; Baek, H.S.; Kim, G.W.; Sundaram, T.; Kang, H.K.; Lee, S.W.; Kim, H.J.; Song, J.K. Human brain activation in response to visual stimulation with rural and urban scenery pictures: A functional magnetic resonance imaging study. Science of the Total Environment. 2010, 408, 2600-2607.

Reviewer 3 Report

This study tried to figure out two factors, i.e. natural vs. built environment, and with vs. without human figure, in the nightscapes that affect the fear perception. It is an interesting study and could be an important paper.

For the research design, this study belongs to “Two-Way Factorial Design”. Thus, the study can actually explore the main effects of nightscape types and human figure that affect fear perception, and as well as the interaction of the two factors.

The “normal” procedure is using “Two-Way ANOVA” to analyze the “whole data” of 40 participants, rather than divided the data into four groups at the beginning. If the two factors have interaction between them, it’s then reasonable dividing the data into different groups to compare them.

The way that the study presented cannot show the main effects of nightscape type and of human figure, neither whether there is an interaction between the two factors. It’s a pity if the study presented in this form because the study design and the data quality already allow a potential of more expressive results.

The same is analyzing the effects on alpha and beta waves which act as other two dependent variables besides fear perception. Two-Way ANOVA should also be adopted.  

The authors can use the original data and reanalyze it again towards fear perception, alpha waves, and beta waves.

Please rewrite your method, the results, and discuss them renewed.

Besides, two small things could be considered:

1. Image (a) and (b) of Fig. 2 seems inverted.

2. Unedified abbreviation should be better not used in the abstract.

Author Response

First of all, we are very grateful for the detailed review of our paper. We believe your comments will greatly contribute to enhancing the completeness of our manuscript, what we modified and additional comments are listed below in order.

Revised parts are marked red in manuscript file.

The “normal” procedure is using “Two-Way ANOVA” to analyze the “whole data” of 40 participants, rather than divided the data into four groups at the beginning. If the two factors have interaction between them, it’s then reasonable dividing the data into different groups to compare them.

-> We appreciate you pointed out very critical part of this study. We reanalyzed data with Two-Way ANOVA in perceived fear differences depending on landscape type and adult presence.

The way that the study presented cannot show the main effects of nightscape type and of human figure, neither whether there is an interaction between the two factors. It’s a pity if the study presented in this form because the study design and the data quality already allow a potential of more expressive results.

-> In carrying out the Two-Way ANOVA, we analyzed the main effect, interaction effect, and post-hoc. Also, the revised result, initial setting, and implication from this were added in every part of manuscript.

The same is analyzing the effects on alpha and beta waves which act as other two dependent variables besides fear perception. Two-Way ANOVA should also be adopted.  

-> We tried Two-Way ANOVA in EEG part like perceived fear part above, but the interaction between factors was not verified.

Round 2

Reviewer 2 Report

There are several issues identified on the previous review that continue unsolved (please see previous review), regarding sample analysis, selected cases and used methods.

Author Response

Comments and Revisions

First of all, we are very grateful for the detailed review of our paper. We believe your comments will greatly contribute to enhancing the completeness of our manuscript, what we added and comments are listed below.

Added and revised parts are marked in red.

* Comment

There are several issues identified on the previous review that continue unsolved (please see previous review), regarding sample analysis, selected cases and used methods.

-> We added new input data to complement the data analysis and conclusion. Please see revised contents, table, and figures in the manuscript. Thank you for your thorough review of our manuscript.
